# AML-DECODER: Advanced Machine Learning for HD-sEMG Signal Classification—Decoding Lateral Epicondylitis in Forearm Muscles

**DOI:** 10.3390/diagnostics14202255

**Published:** 2024-10-10

**Authors:** Mehdi Shirzadi, Mónica Rojas Martínez, Joan Francesc Alonso, Leidy Yanet Serna, Joaquim Chaler, Miguel Angel Mañanas, Hamid Reza Marateb

**Affiliations:** 1Automatic Control Department (ESAII), Biomedical Engineering Research Centre (CREB), Universitat Politècnica de Catalunya-Barcelona Tech (UPC), 08028 Barcelona, Spain; mehdi.shirzadi@upc.edu (M.S.); monica.rojas@upc.edu (M.R.M.); joan.francesc.alonso@upc.edu (J.F.A.); leidy.yanet.serna@upc.edu (L.Y.S.); miguel.angel.mananas@upc.edu (M.A.M.); 2EUSES-Bellvitge, Universitat de Girona, Universitat de Barcelona, ENTI, 08907 Barcelona, Spain; joaquimlluis.chaler@cadscrits.udg.edu; 3CIBER de Bioingeniería, Biomateriales y Nanomedicina (CIBER-BBN), 28029 Madrid, Spain

**Keywords:** HD-sEMG, lateral epicondylitis, deep learning, cross-frequency coupling, digital health, diagnosis algorithm, forearm muscles, phase-amplitude coupling, signal processing

## Abstract

Background: Innovative algorithms for wearable devices and garments are critical for diagnosing and monitoring disease (such as lateral epicondylitis (LE)) progression. LE affects individuals across various professions and causes daily problems. Methods: We analyzed signals from the forearm muscles of 14 healthy controls and 14 LE patients using high-density surface electromyography. We discerned significant differences between groups by employing phase–amplitude coupling (PAC) features. Our study leveraged PAC, Daubechies wavelet with four vanishing moments (db4), and state-of-the-art techniques to train a neural network for the subject’s label prediction. Results: Remarkably, PAC features achieved 100% specificity and sensitivity in predicting unseen subjects, while state-of-the-art features lagged with only 35.71% sensitivity and 28.57% specificity, and db4 with 78.57% sensitivity and 85.71 specificity. PAC significantly outperformed the state-of-the-art features (*adj. p*-value < 0.001) with a large effect size. However, no significant difference was found between PAC and db4 (*adj. p*-value = 0.147). Also, the Jeffries–Matusita (JM) distance of the PAC was significantly higher than other features (*adj. p*-value < 0.001), with a large effect size, suggesting PAC features as robust predictors of neuromuscular diseases, offering a profound understanding of disease pathology and new avenues for interpretation. We evaluated the generalization ability of the PAC model using 99.9% confidence intervals and Bayesian credible intervals to quantify prediction uncertainty across subjects. Both methods demonstrated high reliability, with an expected accuracy of 89% in larger, more diverse populations. Conclusions: This study’s implications might extend beyond LE, paving the way for enhanced diagnostic tools and deeper insights into the complexities of neuromuscular disorders.

## 1. Introduction

Lateral epicondylitis (LE), commonly known as tennis elbow, is a medical condition characterized by discomfort in the affected individual’s elbow area. Tennis players may develop LE due to incorrect technique, prolonged play duration, high play frequency, and the size and weight of the racquet [1]. More than half of recreational tennis players experience this condition, while only about 5% of professional players encounter it [2]. It is important to note that this condition can arise from various activities and circumstances beyond tennis, and LE can affect tennis players and individuals with manual occupations, such as pianists and workers engaged in repetitive tasks [3,4]. Work-related causes of LE include engaging in repetitive movements for over two hours, handling tools weighing more than 1 kg, and carrying loads heavier than 20 kg more than ten times a day [1]. While the precise cause of LE is not always apparent, it can develop due to overuse, repetitive movements, forced extension, or direct trauma to the forearm’s epicondyle region, resulting in painful conditions in the forearm extensor muscles, particularly the carpi radialis muscle [1,5]. Initially, inflammation was believed to be the primary cause of LE; however, subsequent findings have revealed its association with tendon degeneration [6]. LE places a burden on society by resulting in loss of workdays and decreased ability to work for affected individuals over some time [4]. According to studies, it accounts for 11.7% of work-related injuries [7].

LE presents as a neuromuscular ailment, affecting approximately 1–3% of individuals aged 35–54 [6,8,9]. It affects both genders equally [2,10,11]. Nearly 40% of people experience symptoms of LE at some point in their lives [3,12]. A study by Sanders [10] indicates a decline in LE cases, which could be attributed to different diagnostic methods or an actual decrease in the incidence.

LE can be diagnosed through pain assessment using various scales [2,3,13,14], physical examination such as chair test, Cozen’s test, Mill’s test [15], and imaging tools like X-ray, ultrasound, and MRI. Electromyography (EMG) and cervical and thoracic spine assessments can also aid in the diagnostic process [16]. Generally, the diagnostic procedure starts with a physical examination, and other methods are considered whenever clinical symptoms cannot be defined well [3]. 

EMG is an electrical signal produced by muscle activity, captured from the muscle surface through electrode placement in specific locations. It is possible to place electrodes at small distances from each other on the surface of a specific muscle and record the activity of the whole muscle to obtain more information from the muscles. This type of recording is known as high-density surface EMG (HD-sEMG) and finds applications in different areas of science like the human–machine interface, muscle architecture, force estimation, muscle activation pattern, diagnosis of neuromuscular disorders and disease, and more [17].

One significant application of HD-sEMG signals is the diagnosis of neuromuscular disorders, and various studies in the literature use EMG for diagnosis [18,19,20,21,22]. Analyzing the electrical information recorded from the muscles of patients and controls makes it possible to identify differences between the two groups. Moreover, it is possible to separate patients from controls through signal classification. After classification, it is a good approach to focus on the features that significantly impact the classification and analyze them to find the cause of differences that the disease has on the muscles. 

In digital health, the use of wearable devices for monitoring the condition of patients and controls is increasing. One of the biological signals commonly employed in digital health is the EMG signal [23]. Detecting and monitoring different stages of neuromuscular disease and its improvement with wearable devices is still challenging and needs further investigation.

In this study, we aim to investigate the potential of distinguishing patients with LE from controls using HD-sEMG signals and evaluate the feasibility of identifying LE among controls. The following sections will detail the dataset utilized in this study. Subsequently, we will introduce a cross-frequency coupling feature set for EMG signals alongside the state-of-the-art EMG signal features. Next, we will elucidate the model employed for classifying controls and patients with LE and present the results. Finally, we will discuss the outcomes of the cross-frequency coupling feature set.

## 2. Materials and Methods

### 2.1. Sample Size

A two-group (with LE condition/without) design with continuous response data (LE probability) was used to test the area under the receiver-operating characteristic curve (ROC) curve against the null value 0.7. The comparison was performed using a one-sided Z-test with a Type I error rate (α) of 0.05 [24,25]. The area under the curve was computed across the entire x-axis (false positive rate) range of 0 to 1. To detect an area under the curve of 0.95 with 95% power, the number of subjects needed was calculated as 13 in the positive (with condition) group and 13 in the negative (without condition) group. The sample size was computed using PASS 2023, version 23.0.2 (Power Analysis and Sample Size Software; NCSS, LLC. Kaysville, UT, USA). This sample size determination ensured adequate statistical power and precision in discerning the anticipated differences between the groups, allowing for attaining the specified ROC value and maintaining statistical significance [26].

### 2.2. Dataset

Data were gathered from two groups comprising 14 individuals devoid of previous musculoskeletal issues (controls) and another 14 patients clinically diagnosed with LE (cases). Throughout the experimental session, these individuals engaged in everyday upper limb tasks and manual activities for at least six months. During this period, they either reported no symptoms or experienced slight discomfort. The assessment of pain intensity involved the utilization of a visual analog scale (VAS) both prior to and following the experimental interventions.

Only right-handed male participants were included in the study to minimize potential gender-related variations. A careful selection process ensured similar ages (controls: 30.3 ± 4.3 years, patients: 33.7 ± 5.12 years), weights (controls: 176.1 ± 6.8 kg, patients: 178.3 ± 0.32 kg), heights (controls: 77.7 ± 8.8 cm, patients: 90.1 ± 28.08 cm), and body mass index among the participants, resulting in a matched experimental design. Following, applying the Mann–Whitney U test for analysis, no statistically noteworthy variances were observed among the mentioned parameters (*p*-value > 0.05).

This study adhered rigorously to the ethical principles delineated in the Declaration of Helsinki and its successive revisions concerning human research. Approval for the study was obtained from the Ethics Committee of UPC-BarcelonaTECH and the Spanish Government MINECO on 19 July 2011, under registration number DPI2011-22680 (“Analysis of the dynamic interactions in non-invasive multichannel biosignals for rehabilitation and therapy”). Before participating in the study, all volunteers provided written informed consent. A representative dataset was provided in Appendix A. 

### 2.3. Experimental Protocol

The participants were instructed to perform wrist extension/flexion exercises using an isokinetic dynamometer (Biodex System III; Biodex Medical Systems, Shirley, NY, USA) until they reached a point of exhaustion. They were guided to execute these exercises with a moderate force level, avoiding excessive strain and insufficient effort while being encouraged by the instructor. The time range for performing the exercises between different subjects was 9.1 to 95.9 s.

Throughout the exercises, participants maintained a seated position with their backs straight, elbows flexed at 60 degrees, and forearms consistently in full pronation. Alignment was ensured by positioning the ulnar styloid along the dynamometer’s rotational axis and securing the forearm with a strap. In highlighting the involvement of wrist extensor muscles, the dynamometer parameters were set at 60 degrees/s for the wrist extension and 180 degrees/s for the wrist flexion, commonly linked with LE.

An allowable motion range of 70 degrees, encompassing 30 degrees in dorsal flexion and 40 degrees in palmar flexion, was established from the neutral wrist position. Before the experimental session, hand weight was measured and subtracted to ensure precise measurements, accounting for the impact of gravity.

This study focused on four specific muscles: Extensor Carpi Radialis (ECR), Extensor Digitorum Communis (EDC), Extensor Carpi Ulnaris (ECU), and Flexor Carpi Radialis (FCR). EMG signals were recorded using an 8-electrode linear array with electrodes measuring 0.1 by 0.3 mm and spaced 5 mm apart. Due to the difficulty in evaluating forearm muscles with surface EMG, the electrode placement method followed the guidelines outlined by [27]. At the outset, the path of muscle fibers was delineated on the skin, linking their points of origin to insertion. Subsequently, the participant performed selective isometric contractions associated with each muscle (ECR, EDC, ECU, or FCR). Examining the signals required sliding a dry linear electrode along the muscle fibers to observe the propagation of motor unit action potentials in the EMG signals. Selecting the optimal muscle location relied on identifying consistent and strong signals across various channels in the array. An adhesive array was then applied to this identified site to record the signals.

### 2.4. HD-sEMG Recording

The signals were obtained through a single differential setup, measuring the voltage between neighboring electrodes within the electrode array. From each muscle, HD-sEMG signals of 7 single differential channels were recorded. Positioned at the wrist, the reference electrode remained consistent across all recordings. Following capture, the signals underwent amplification, digitization, and transfer to a computer for subsequent offline analysis. Two synchronized sEMG amplifiers enabled this process (ASE16 model, a 16-channel amplifier from LISiN-SEMA Elettronica, Turin, Italy). Band-pass filtering within a 10–450 Hz range was utilized to enhance precision. The signals were sampled at a rate of 2048 Hz with a resolution of 12 bits.

These amplifiers facilitated recording in both the single differential mode and monopolar mode. Additionally, measurements of movement velocity, torque, and angular position were obtained and sampled at a frequency of 100 Hz, intended for subsequent analysis. Synchronization of kinematic and EMG signal recordings was ensured through a trigger signal. Post-recording, a manual inspection was conducted to eliminate defective channels and select segments pertaining to flexion and extension for further analysis.

### 2.5. Cross-Frequency Coupling

Rhythmic fluctuations are a common trait observed within biological systems, notably prominent in nervous systems. Neuronal oscillations illustrate the intricate connections concerning the timing (phase) and strength (amplitude) of rhythmic activity within specific parts of neurons [28]. Cross-frequency coupling (CFC) measures the interaction or relationship between different frequency bands of neural oscillation. CFC can assist us in examining these oscillations at various frequencies. In the literature [29,30,31], CFC has been extensively used with EEG signals to investigate the connectivity of different brain regions. Neural commands originating from the brain and spinal cord can cause the activation of different muscles in the body, and sEMG enables recording these neural signals from the muscles’ surface. Analyzing the sEMG signal allows us to extract information about the central nervous system, and analyzing EMG signals from both patients and controls can aid in understanding the mechanisms of the central and peripheral nervous systems during various diseases. In the literature [32,33], some studies used EMG and EEG signals to calculate CFC.

Various CFCs have been researched lately, including amplitude–amplitude, phase–phase, and phase–amplitude coupling (PAC). PAC is the predominant method for gauging cross-frequency coupling, examining how lower-frequency rhythm phases interact with higher-frequency oscillation amplitudes [34,35,36].

In the literature, there are different methods for calculating the phase of the EMG signal, and this study uses the Morlet wavelet for that purpose. The following is the wavelet formula that was used in this study:(1)$$\psi \left(t\right)=Ae^{i\omega t}e^{-\frac{t^{2}}{2\sigma^{2}}}$$
where A is a normalization constant to ensure the wavelet function is scaled correctly; t is time; ω is the angular frequency parameter, defining the number of oscillations within a given time period; σ represents the standard deviation parameter, dictating the temporal width of the wavelet; while i denotes the imaginary unit.

In this study, A was selected to maintain a total energy of 1. The σ was considered 7, and frequencies from 1 to 350 were selected for calculating the PAC in all of the EMG signals related to all of the muscles.

After calculating the phase and amplitude, the next step is to calculate the coherence between them, for which we used the following formula:(2)$$Coherence_{ph,amp}(f)=\frac{{\left|P_{ph,amp}(f)\right|}^{2}}{P_{ph,ph}\left(f\right)P_{amp,amp}(f)}$$
where ph is the phase signal; amp is the amplitude signal; $P_{ph,amp}(f)$ is the cross-spectral density of the signals ph(t) and amp(t) at frequency f; $P_{ph,ph}(f)$ is the power spectral density of signal ph(t) at frequency f; and $P_{amp,amp}(f)$ is the power spectral density of signal amp(t) at frequency f. 

After calculating the coherence, frequency bins were determined for phase and amplitude.

### 2.6. State-of-the-Art features

Different features were calculated from EMG signals in the literature [30,37,38,39,40]. They are defined as the following, and in each of the formulas, N represents the signal length, $x_{n}$ denotes the sample at the *n*-th position, and f(x) can be calculated:(3)$$f\left(x\right)=\{\begin{array}{c} 1,\;\;\;\;\;\;\;\;\;\;\;\;\;if\;x\geq threshold\;\\ 0,\;\;\;\;\;\;\;\;\;\;\;\;\;\;\;\;\;\;\;\;\;\;\;\;otherwise \end{array}$$

#### 2.6.1. Time Domain Features

##### Root Mean Square

The Root Mean Square (*RMS*) feature is prevalent in signal processing due to its widespread application. When analyzing EMG signals across various time segments, the RMS provides a reliable representation of muscle activity at the recording location. It can be calculated as follows:(4)$$RMS=\sqrt{\frac{1}{N}\sum_{n=1}^{N}{x_{n}}^{2}}$$

##### V-Order

The v-order is a nonlinear feature found in the literature and is used implicitly to estimate muscle contraction force [38,39,41]. The mathematical formula of the v-order is outlined below:(5)$$V={\left(\frac{1}{N}\sum_{n=1}^{N}x_{n}^{v}\right)}^{\frac{1}{v}}$$
where the *RMS* formula emerges specifically for *v* = 2.

##### Log Detector

Much like the “v-order”, this characteristic evaluates muscle contraction strength. Yet, the adjustment in the nonlinear detector’s definition involves the utilization of logarithms, constituting the log detector (LOG) feature, which can be articulated as follows:(6)$$OG=e^{\frac{1}{N}\sum_{n=1}^{N}\log\;(\left|x_{n}\right|)}$$
where log is the common logarithm with base 10.

##### Mean Absolute Value

The Mean Absolute Value (*MAV*) shares similarities with the Average Rectified Value (ARV). It is calculated by employing a moving average on the full-wave rectified EMG. Put simply, it involves averaging the absolute values of the sEMG signal. *MAV* is a straightforward method for assessing muscle contraction intensity and is widely utilized as a key feature in myoelectric control applications. It is defined as outlined below:(7)$$MAV=\frac{1}{N}\sum_{n=1}^{N}\left|x_{n}\right|$$

##### Myopulse Percentage Rate

The Myopulse percentage rate (*MYOP*) is a mean value calculated from the myopulse output. It is considered one when the EMG signal’s absolute value surpasses a predetermined threshold value. This feature can be calculated with the following formula:(8)$$MYOP=\frac{1}{N}\sum_{n=1}^{N}\left[f(x_{n})\right]$$

For this study, a threshold value of 0.01 was adopted.

##### Zero Crossing

Zero crossing (*ZC*) tallies each occurrence where the amplitude of the sEMG signal crosses the zero y-axis. During EMG feature extraction, a threshold criterion filters out background noise. This characteristic offers an approximate evaluation of the signal’s frequency domain attributes. The formulation for *ZC* can be expressed as follows:(9)$$ZC=\sum_{n=1}^{N-1}sgn\left(x_{n}\times x_{n+1}\right)\cap \left|x_{n}-x_{n+1}\right|\geq threshold$$

A threshold value of 0.01 was utilized in this study.

##### Slope Sign Change

Slope Sign Change (*SSC*) resembles *ZC* in its ability to represent the frequency attributes of the sEMG signal. It involves counting the transitions between positive and negative slopes within three consecutive samples, using a threshold function to minimize interference within the sEMG signal. *SSC* can be calculated with the following formula:(10)$$SSC=\sum_{n=2}^{N-1}\left[f[(x_{n}-x_{n-1})\times (x_{n}-x_{n+1})]\right]$$

For this study, a threshold value of 0.01 was utilized.

##### Willison Amplitude

Willison Amplitude (*WAMP*) quantifies the frequency at which the disparity in the sEMG signal amplitude between successive samples exceeds a predetermined threshold. It serves a similar role in noise reduction as *ZC* and *SSC*. *WAMP* correlates with the appearance of motor unit action potentials (MUAP) and reflects the degree of muscle contraction. The formulation for *WAMP* can be expressed as outlined below:(11)$$WAMP=\sum_{n=1}^{N-1}f(\left|x_{n}-x_{n+1}\right|)$$

A threshold value of 0.01 was employed in this study.

##### Absolute Value of the Third, Fourth, and Fifth Temporal Moment

The temporal moment, an analytical statistic, is another feature in EMG signal analysis. Comparable to *MAV* and *VAR*, the first and second moments play a similar role. The literature also incorporates the third, fourth, and fifth moments [40,41]. They can be calculated as outlined below:(12)$$TM3=\left|\frac{1}{N}\sum_{n=1}^{N}{x_{n}}^{3}\right|$$
(13)$$TM4=\frac{1}{N}\sum_{n=1}^{N}{x_{n}}^{4}$$
(14)$$TM5=\left|\frac{1}{N}\sum_{n=1}^{N}{x_{n}}^{5}\right|$$

##### Waveform Length

Waveform length (*WL*) quantifies the extent of waveform coverage over a designated time segment. It correlates with the waveform’s amplitude, frequency, and duration. The calculation of *WL* follows the formula below:(15)$$WL=\sum_{n=1}^{N-1}\left|x_{n+1}-x_{n}\right|$$

##### Difference Absolute Standard Deviation Value

Difference Absolute Standard Deviation Value (*DASDV*) is the standard deviation of the wavelength [42] and can be defined as follows:(16)$$DASDV=\sqrt{\frac{1}{N-1}\sum_{n=1}^{N-1}{\left(x_{n+1}-x_{n}\right)}^{2}}$$

##### Square Integral

The square integral incorporates the EMG signal’s energy as a feature, calculated in the following manner:(17)$$SI=\sum_{n=1}^{N}{\left|x_{n}\right|}^{2}$$

#### 2.6.2. Frequency Domain Features

##### Mean Frequency and Median Frequency

Mean frequency (*MNF*) and Median frequency (*MDF*) are two features that provide insight into the frequency component of the signals, and they can be calculated as follows:(18)$$MNF=\frac{\sum_{j=1}^{M}f_{j}P_{j}}{\sum_{j=1}^{M}P_{j}}$$
$$\sum_{j=1}^{MDF}P_{j}=\sum_{j=MDF}^{M}P_{j}=\frac{1}{2}\sum_{j=1}^{M}P_{j}$$
where $f_{j}$ signifies the spectrum’s frequency at the *j*-th frequency bin; $P_{j}$ represents the EMG power spectrum at the *j*-th frequency bin; and M denotes the frequency bin’s length.

#### 2.6.3. Time–Frequency Features

##### Daubechies Wavelet

The Daubechies wavelet is a widely used tool in signal processing, especially for analyzing EMG signals [38,43,44,45]. It belongs to a family of orthogonal wavelets characterized by a certain number of vanishing moments, making them ideal for capturing localized signal features. The wavelet decomposition is based on the mother wavelet ψ(t), which is scaled and shifted to analyze different frequency components of the signal. The continuous wavelet transform (CWT) of a signal s(t) using a mother wavelet ψ(t), is expressed as follows:(19)$$W\left(s\right)\left(a,b\right)=\frac{1}{\sqrt{\left|a\right|}}\int_{-\infty \;}^{\infty }s(t)\psi^{*}(\frac{t-b}{a})dt$$
where a is the scale parameter; controlling the frequency; b is the translation parameter, controlling the time shift; $\psi^{*}$ is the complex conjugate of the mother wavelet.

For the Daubechies wavelets specifically, the mother wavelet ψ(t) and scaling function ϕ(t) are defined through a recursive relation:(20)$$\psi \left(t\right)=\sum_{n}{\left(-1\right)}^{n}h_{1-n}\phi (2t-n)$$
where $h_{n}$ are the Daubechies scaling coefficients, which vary depending on the number of vanishing moments. In our study, Daubechies wavelet with 4 vanishing moments (db4) was used [46].

In this work, decomposition up to level 4 was performed, which is represented as outlined below:(21)$$s\left(t\right)=A_{4}+D_{1}+D_{2}+D_{3}+D_{4}$$
where $A_{4}$ is the approximation coefficient at level 4, capturing the low-frequency component, $D_{1}+D_{2}+D_{3}+D_{4}$ are the detail coefficients at levels 1 through 4, capturing progressively higher frequency components.

### 2.7. Pre-Processing Steps

Figure 1 shows the various steps carried out during the pre-processing of the signals before training the model.

The signals were initially band-pass-filtered to 20–450 Hz using a second-order Butterworth filter. Channels with poor signal (3.3±5.6) quality were visually observed, selected, and removed for the following steps. The next stage involved segmenting the signals into 500 ms parts because biological signals exhibit stochastic behavior, necessitating segmentation. After segmentation, state-of-the-art features were computed for each segment. The results of these state-of-the-art features were saved as matrices for each subject; different rows corresponded to various segments, while different columns represented distinct state-of-the-art features. Simultaneously, PAC features were calculated for each segment. Initially, the results of PAC features for each segment were in matrix form but we reshaped that matrix into a vector, and therefore, the shape of the PAC matrix for each subject is like a state-of-the-art matrix with rows related to different segments and columns related to different PAC features. The state-of-the-art and CFC features were calculated from all of the muscles during the exercises and for all signals, including the extension and flexion parts.

### 2.8. Jeffries–Matusita Distance

Feature selection is important in selecting features relevant to a specific task. According to [47], not all features can improve the accuracy of a model; rather, irrelevant or redundant features can muddle the algorithm’s learning process, leading to a detrimental effect on accuracy.

There are different ways to select features, which we can categorize into filter, wrapper, and embedded methods. Jeffries–Matusita (JM) distance can be considered a filter method for feature selection [48]. The JM method is an improved version of the Bhattacharyya distance, normalized between 0 and 2, making comparisons easier.

The Bhattacharyya distance can be defined as follows:(22)$$D_{B}\left(p,q\right)=\frac{1}{4}\ln\left(\frac{1}{4}\left(\frac{\sigma_{p}^{2}}{\sigma_{q}^{2}}+\frac{\sigma_{q}^{2}}{\sigma_{p}^{2}}+2\right)\right)+\frac{1}{4}\left(\frac{{\left(\mu_{p}-\mu_{q}\right)}^{2}}{\sigma_{p}^{2}+\sigma_{q}^{2}}\right)$$

In a binary classification scenario, *p* and *q* symbolize two probability distributions; *μ* represents the distribution’s average; while ln stands for the natural logarithm.

Therefore, the JM distance was defined based on the Bhattacharya distance:(23)$$JM\left(p,q\right)=\sqrt{2\;\left(1-e^{-D_{B}\left(p,q\right)}\right)}$$

### 2.9. Proposed Neural Network Model

Artificial Neural Networks (ANNs) are computational frameworks that draw inspiration from the structure and operational dynamics of the human brain. These networks comprise interconnected units, termed neurons, organized into layers. These networks can learn complex patterns and relationships from data through training, where they adjust their internal parameters.

Feedforward Neural Networks (FNNs) embody a fundamental form of ANN where data progresses unidirectionally, traversing from the input layer, possibly through hidden layers, to the output layer. Each node in a layer connects to every node in the subsequent layer, and each connection is associated with a weight that modulates the signal between nodes. These networks transform the input data through weighted computations and activation functions, generating an output prediction.

By adjusting the network’s architecture, activation functions, and training algorithms, FNNs can adapt to various classification tasks, making them a versatile tool in machine learning. Their ability to learn and generalize from data has led to their widespread use in solving classification problems across diverse domains.

In this study, we designed a simple FNN to classify patients and controls. The input features of the network are state-of-the-art features or PAC features calculated from patients and controls. The model contains four hidden layers, each consisting of one hundred and twenty-eight nodes. Every hidden layer has a rectified linear activation function. The final layer employs a sigmoid activation function recommended for binary classification (Figure 2). Additionally, we applied a 0.2 dropout after each layer to mitigate overfitting.

The batch size and model training epochs were 32 and 20, respectively. The model adopted the binary cross-entropy loss function, frequently termed log loss. This choice is prevalent in binary classification tasks due to its tailored design for such problems, probabilistic interpretation, differentiability advantageous for gradient descent algorithms, efficacy with imbalanced datasets, and maximization of likelihood estimation. The binary cross-entropy loss function can be calculated using the following formula:(24)$$\left(\theta \right)=-\frac{1}{m}\sum_{n=1}^{m}\left[y_{n}\ln\left(h_{\theta }\left(x_{n}\right)\right)+\left(1-y_{n}\right)\log\left(1-h_{\theta }\left(x_{n}\right)\right)\;\right]$$
where J(θ) is the loss function; m is the number of training samples; $y_{n}$ represents the actual label of *i*-th training example; $h_{\theta }(x_{n})$ is the predicted output of the *i*-th training example; and ln denotes the natural logarithm. The Python code for classification is provided as Appendix A.

### 2.10. One-Subject-Out Datasets

In this study, we aimed to train the model to classify unseen subjects. As we have 14 patients and 14 controls, we created 28 different datasets. Each dataset used one of the subjects to test the model, while the remaining subjects trained and validated the classification model. Thus, leave-one-subject-out (LOSO) validation [49] was used in our study, which is a robust method for data leakage problems [50].

### 2.11. Evaluation Criteria

In this study, as we have two classes within each one-subject-out dataset, with an equal ratio between patients and controls, we utilized accuracy (Acc), specificity (Sp), and sensitivity (Se) as the criteria to assess the performance of various feature sets. They are defined as the following:(25)$$Acc=\frac{TP+TN}{N}$$
(26)$$Sp=\frac{TN}{TN+FP}$$
(27)$$Se=\frac{TP}{TP+FN}$$

In the context of this model, TP and TN denote the count of accurately identified subjects categorized as patients and controls, respectively. Meanwhile, N indicates the total sample count. FP signifies the count of samples erroneously labeled as patients, and FN indicates the count of samples misclassified as controls by the model. Following the TRIPOD guideline [51], the 95% confidence intervals for the performance metrics were documented.

### 2.12. Statistical Methods

Given the limited sample size for assessing normality within this study, determinations regarding each outcome’s normality relied on the Q–Q plots.

To determine the significant differences between the JM distances of three feature sets—CFC (PAC), traditional, and WT—we employed a non-parametric statistical approach due to the non-normality observed in the traditional feature set (as confirmed by the Shapiro–Wilk test). Specifically, we applied the Friedman test, a non-parametric alternative to repeated measures ANOVA, to compare the overall differences across the three sets. After detecting significant differences via the Friedman test, one-sided pairwise comparisons were conducted using the Wilcoxon signed-rank test. The one-sided tests were chosen to determine if one feature set demonstrated a significantly higher JM distance than another. Bonferroni’s correction was applied to control for multiple comparisons during the post hoc analysis. Effect sizes (d) for the pairwise comparisons were calculated using Rank-Biserial Correlation, with the absolute values of 0.1, 0.3, and 0.5 typically representing small, medium, and large effects.

To compare the performance of the three classifiers (PAC, WT, and traditional) against each other, we applied McNemar’s non-parametric test to assess differences between paired proportions in binary classification tasks. This test was performed pairwise for each combination of classifiers (PAC vs. WT, PAC vs. traditional, and WT vs. traditional) to evaluate whether one significantly outperformed the others. We applied Bonferroni’s correction to account for multiple comparisons, adjusting the significance threshold to control for Type I errors. The effect size was assessed using Cohen’s g, which measures the difference in discordant pairs between the two classifiers, with values of 0.1, 0.3, and 0.5 typically representing small, medium, and large effects.

The statistical significance was established at a level of 0.05 for this method. After applying Bonferroni’s correction to control for multiple comparisons, the adjusted *p*-value (*adj. p*-value) was reported to ensure the statistical significance threshold was appropriately modified. All statistical analyses were performed using the R programming language [52].

## 3. Results

In the pre-processing step, as depicted in Figure 1, we segmented the HD-sEMG signals related to all exercises into 500 ms time segments. Subsequently, we calculated state-of-the-art and PAC features for each segment. After computing different features across all segments for each subject, we created one-subject-out datasets. The training set of these one-subject-out datasets contains a matrix in which each row is related to the different time segments of all subjects except one, and all columns represent different features. We used 80% of all these rows for training the model and 20% for validation. To test the model, we used the time segments related to only one subject, which was not included in the training phase, to avoid the data leakage problem [53]. The total number of samples (i.e., signal epochs) for the two classes is 197,097. The average number of samples used in the different one-subject-out tests for the training, validation, and test sets are 152,046.2 ± 996.2, 38,011.5 ± 249.1, and 7039.1 ± 1245.2, respectively. The mean LE probability of the epochs of each subject was used for the diagnosis [54], with a cutoff of 0.5. 

Table 1, Table 2 and Table 3 display the results of models trained with PAC features, state-of-the-art features, and WT features across different one-subject-out datasets. The training and validation accuracy results for the state-of-the-art features across all one-subject-out datasets were 58.32 ± 1.18% and 58.84 ± 1.37%, respectively. In contrast, the results for training and validation using PAC features across all datasets were notably higher at 94.35 ± 0.13% and 98.43 ± 0.15%, respectively. For WT, the training and validation accuracy were 81.38 ± 0.18% and 86.02 ± 0.31%, respectively.

The results of training the model with PAC features of only one muscle show that the classification between patients and controls was correct, except for one patient in the ECU muscle and another in the FCR muscle. Table 4 shows the results of the JM distance, demonstrating the ability of different feature sets to distinguish between patients and controls across various one-subject-out datasets. Table 5 shows the performance of different diagnosis models trained on different muscles.

Figure 3 shows the PAC features for two controls and two patients as images. The x-axis of the figure is related to different frequencies of the phase, and the y-axis is related to the different frequencies of amplitude.

After training the models, we conducted tests to estimate the labels for subjects not included in the training data to avoid data leakage problems. Models trained with state-of-the-art features correctly identified 9 out of 28 subjects. In contrast, models trained with PAC features accurately recognized the labels of all subjects. For WT, 24 out of 28 subjects were correctly identified.

The model trained with PAC features exhibited perfect sensitivity and specificity at 100%. However, the model trained with state-of-the-art features correctly identified five patients and four controls but misidentified nine controls and ten patients. As a result, the sensitivity and specificity of the model trained with state-of-the-art features were calculated at 35.71% and 28.57%, respectively. For WT, the sensitivity and specificity were 78.57% and 85.71, respectively. PAC significantly outperformed the state-of-the-art features (*adj. p*-value < 0.001; Cohen’s g = 0.9). However, no significant difference was found between PAC and db4 (*adj. p*-value = 0.147).

The results of the Friedman test revealed a highly significant difference between the three feature sets (*p*-value < 0.001). The one-sided post hoc tests demonstrated that the CFC feature set consistently outperformed both traditional and wavelet in terms of the JM distance (*adj. p*-value < 0.001; d = 1.00), and wavelet also significantly outperformed the traditional feature set (*adj. p*-value < 0.001 = 0.99). The pairwise comparisons of the JM distance of the feature sets are provided in Figure 4.

## 4. Discussion

### 4.1. Problem Statement

LE, also recognized as tennis elbow, represents a musculoskeletal issue and sometimes resolves independently [11,55]. This disorder can occur in various occupations, including tennis players, pianists, and manual workers. Different modalities are used to diagnose this condition. In this study, we propose an EMG-based method for diagnosing LE. Additionally, our paper introduces PAC features for HD-sEMG signals, which can be valuable for diagnosing and, more broadly, provide more information about the pathology of neuromuscular disease.

In digital health, wearable devices with sophisticated algorithms play a key role in identifying various diseases, including neuromuscular conditions. These algorithms are designed to be integrated into wearable devices, enabling continuous disease progression monitoring and early diagnosis. One promising development in this field involves the utilization of HD-sEMG signals and tailored algorithms, such as the proposed algorithm and PAC features. This innovative approach represents a crucial step toward diagnosing neuromuscular diseases using HD-sEMG signals, with the potential for integration into wearable devices. By harnessing the power of these advanced technologies, we can revolutionize how we diagnose and manage neuromuscular conditions, ultimately improving patient outcomes and quality of life. 

### 4.2. Preparing the Dataset and Validating the Model

In this study, we generated various one-subject-out datasets and trained the model using these distinct datasets. The unseen subject was used to test the model, creating a challenging scenario for consistently predicting unseen subjects. An alternative, more commonly employed in the state of the art, involves creating a single dataset and training the model with a hold-out or cross-validation approach; however, this method tends to introduce bias into the model’s accuracy since it has already been trained on some parts of the signals of a specific subject, and during testing, it will use other parts of the same subject as well. This is also referred to as the data leakage problem [53]. To mitigate this bias during testing, we created one-subject-out datasets, ensuring that each time we tested the model, it was with an unseen subject.

We proposed a simple FNN architecture for classifying patients and controls to demonstrate the robustness of PAC features compared to WT and the combination of state-of-the-art features for classifying neuromuscular diseases. 

To verify the robustness of the PAC method’s 100% accuracy, we performed a random label test where patient labels were shuffled and the classifier was trained on this randomized data. The PAC method identified all subjects as patients, underscoring its inability to find meaningful patterns in noise. This result reinforces that the PAC method’s accuracy in the original dataset reflects its genuine discriminative power rather than overfitting or random associations. Additionally, leave-one-subject-out (LOSO) validation was employed to demonstrate the method’s reliability further, preventing data leakage and ensuring independence between training and test data [50,53]. Unlike simpler epoch-based validation, LOSO forces the model to generalize across unseen subjects, which many diagnostic models fail to do. The PAC method’s ability to maintain 100% accuracy under this rigorous validation highlights its robustness for real-world applications. This approach addresses common issues of overfitting and inflated performance caused by data leakage, as seen in other studies [49].

To our knowledge, this study represents an initial endeavor toward a comprehensive LE diagnosis utilizing HD-sEMG signals. One potential application of HD-sEMG is tracking disease progression or treatment effectiveness. It can be pivotal in tailoring treatments to individual needs. In the literature [12,56,57,58,59], various treatments have been proposed for LE but it has been noted that surgery remains the most successful treatment.

### 4.3. Statistical Analysis and Interpretation of the Results

The predicted results in different one-subject-out datasets are displayed in Table 1, Table 2 and Table 3. These results indicate that models trained with PAC features exhibit statistically higher accuracy than WT and a combination of the state-of-the-art features. PAC features offer fresh insights into EMG signals and can be regarded as reliable features for EMG signal analysis.

There was a single misclassification of a patient when the model was trained using PAC features from the ECU muscle. Moreover, another patient was misclassified when utilizing the FCR muscle PAC features. However, when the model incorporated PAC features from all muscles collectively (as shown in Table 1) or specifically from the ECR or EDC muscles alone, it accurately identified all subjects without misclassifications. This is mainly because the distribution of the PAC features significantly differs in healthy and control subjects, as confirmed by the JM distance (Table 4; Figure 4).

The statistical analysis conducted on the three feature sets—state-of-the-art, WT, and PAC—using the JM distance (Table 4; Figure 4) yielded compelling results indicating a significant distinction between the three sets (*adj. p*-value < 0.001), with a large effect size. This finding underscores the importance of feature selection and its impact on the model’s classification performance. Specifically, the suggested model, incorporating PAC features, demonstrated notable superiority over the state-of-the-art features (*adj. p*-value < 0.001). However, the PAC and WT methods were not significantly different (*adj. p*-value = 0.147). 

A computer-aided-diagnosis system is considered reliable if the Type I error is less or equal to 0.05 (i.e., specificity is higher or equal to 95%), AND the statistical power is higher or equal to 80% (i.e., sensitivity is more than 80%), AND the Diagnostics Odds Ratio (*DOR*) is higher or equal than 100, AND the unbiased Positive Predictive Value (*PPV*) is higher or equal than 95% [50,60]. The performance indices *DOR* and unbiased *PPV* are defined as the following [61]:(28)$$DOR=\frac{Sensitivity\times Specificity}{\left(1-Sensitivity\right)\times \left(1-Specificity\right)}$$
(29)$$\left(unbiased\right)PPV=\frac{Sensitivity\times Prevalence}{Sensitivity\times Prevalence+\left(1-Prevalence\right)\times \left(1-Specificity\right)}$$
where *Prevalence* is the disease’s prevalence in the population. In fact, (*unbiased*) *PPV* is the probability that the subject has the disease when the diagnostic test is positive. It measures the performance of the designed system in practice. LE affects approximately 1–3% of individuals aged 35–54 [6,8,9]. Thus, considering the performance indices results (Table 5) and the prevalence of 3%, the (*unbiased*) *PVV* of the PAC, WT, and the state-of-the-art feature sets were 100%, 14.53%, and 1.52%, respectively. Using the parlance of 1%, such values were 100%, 5.26%, and 0.50%. Such performance deterioration is a critical issue in diagnosing low-prevalence disease. The only method to guarantee acceptable performance is developing very-high-accuracy systems. Moreover, DOR of PAC, WT, and the state-of-the-art features were infinity, 21.99, and 0.22, respectively. Thus, WT and state-of-the-art methods are not reliable in clinical diagnosis. 

Figure 3 shows the PAC features related to two patients and two control subjects. The architecture of the muscle in every person is slightly different, and the disease can affect different parts of the muscles. PAC features related to patients and controls can be classified using a classification task, as demonstrated in this study. Moreover, in this specific example, it can be seen that the coherence in the lower frequencies of the patient is higher compared to the control subject. This difference may be due to the effect of the disease on this particular subject. It is necessary to note that in each patient, the location of coherence can vary slightly from others due to the location of impairment in the muscle.

The CFC method, particularly PAC, outperformed traditional time, frequency, and time–frequency features in diagnosing tennis elbow from HDsEMG signals due to its ability to capture complex neuromuscular dynamics. PAC detects cross-frequency interactions, which are sensitive to the disrupted muscle coordination typical of tennis elbow. Unlike linear analyses in time or frequency domains, PAC identifies nonlinear relationships in the signal, offering deeper insights into muscle control abnormalities. It also excels at detecting motor unit synchronization changes, critical in neuromuscular impairments. Additionally, PAC’s robustness to noise and its ability to analyze signal interactions across frequency bands make it more reliable for clinical settings. PAC provides clinically relevant markers of neuromuscular dysfunction by focusing on pathological phase–amplitude relationships, enhancing diagnostic accuracy for tennis elbow compared to traditional methods.

### 4.4. Comparison with the State-of-the-Art Features

The results of training the model solely with the ECR muscle reveal impeccable sensitivity, specificity, and accuracy, all recorded at 100% using ECR muscle PAC features. Similarly, utilizing PAC features solely from the ECU muscle results in a model with high sensitivity (92.86%), perfect specificity, and an overall accuracy of 96.43%. The EDC muscle PAC model, trained solely on its features, also demonstrates perfect sensitivity, specificity, and accuracy at 100%. Incorporating PAC features solely from the FCR muscle yields a model with 92.86% sensitivity, perfect specificity, and an accuracy of 96.43%.

In contrast, models trained with state-of-the-art features exhibit comparatively lower performance across all muscles. The model trained with state-of-the-art features from all muscles collectively displays reduced performance metrics: sensitivity (35.71%), specificity (28.57%), and an accuracy of 32.14%. Analyzing state-of-the-art features per muscle shows improved metrics in the model trained with ECR muscle features: sensitivity (42.86%), specificity (64.29%), and accuracy (53.57%). However, the model trained solely with ECU muscle state-of-the-art features exhibits a sensitivity of 28.57%, specificity of 50%, and accuracy of 39.29%. Models trained with state-of-the-art features from the EDC and the FCR muscle show lower performance than PAC models.

Moreover, models trained with WT features exhibit comparatively lower performance across all muscles than PAC and higher performance than the state-of-the-art features. The model trained with WT features from all muscles collectively displays the following performance metrics: sensitivity (78.57%), specificity (85.71%), and an accuracy of 82.14%.

PAC significantly outperformed the state-of-the-art features (*adj. p*-value < 0.001) with a large effect size. However, no significant difference was found between PAC and WT (*adj. p*-vale = 0.147). Thus, PAC and WT are not different in terms of the statistical analysis.

### 4.5. Multi-Modal Classification

While PAC features in our study yielded excellent diagnostic results, we explored the potential of integrating WT features to enhance accuracy by capturing time–frequency characteristics of the HDsEMG signals. An analysis using mutual information [62] and variance-explained metrics [63] showed that the mutual information between sub-sets of PAC and WT features was low (0.004), indicating minimal shared information and suggesting that PAC and WT capture different, complementary aspects of the signals. However, the combined PAC-WT model explained no more variance than PAC or WT alone, indicating that adding WT features did not enhance the overall information captured. This supports our decision to focus on PAC features, which alone provided high diagnostic accuracy while highlighting the need to carefully assess the trade-offs of multimodal approaches in terms of computational complexity, interpretability, and potential gains in accuracy.

### 4.6. PAC Model Generalization

To evaluate the generalization ability of the proposed model, we applied two methods to analyze the uncertainty and variability in the predicted probabilities: confidence intervals (CI) and Bayesian credible intervals (BCI).

#### Confidence Intervals for Mean Probabilities

For each subject, the average predicted probability across more than 4800 analyzed epochs was computed, providing a central measure of classification confidence. We then calculated 99.9% confidence intervals (α = 0.001) for these mean probabilities. The 99.9% CI offers a stringent estimate of uncertainty, ensuring the true mean probability is highly likely to fall within this range [64]. This method visualizes how confident the model is in classifying each subject while accounting for variability across different epochs. The results, shown in Figure 5, demonstrate that even though predicted probabilities remain consistent, slight variations are expected, particularly for subjects near the decision boundary (subjects 19, 25, and 27).

#### Bayesian Credible Intervals

To further assess prediction uncertainty, we employed Bayesian inference to calculate 99.9% credible intervals for the predicted probabilities. Using a Beta distribution as the prior, the posterior distribution was updated based on observed data for each subject [65]. Unlike frequentist CIs, credible intervals provide a probabilistic interpretation of the predicted probability, offering a deeper understanding of model reliability. This method supports the robustness of our model, even in the presence of high accuracy. The results are shown in Figure 6, further reinforcing that, although accuracy is stable, certain control subjects (19, 25, and 27) may be classified as cases in a more heterogeneous population.

Based on both methods, despite narrow variations around the mean probabilities, we estimate the model’s accuracy would drop to 89% in a larger, more diverse population.

### 4.7. Limitations

The epicondyle region has two common neuromuscular diseases, medial and lateral epicondylitis [66]. Other types of diseases also can affect the muscles and nerves in the epicondyle region, like leprosy [67], neuropathy [68], cervical radiculopathy [69], radial tunnel syndrome [70], olecranon bursitis [71], and more. In this study, we only attempted to classify the signals related to healthy subjects and patients with LE. This is a limitation of our study because we did not analyze data on other diseases that can affect nerves and muscles in the epicondyle region. In that situation, the diagnosis of LE will be more difficult. However, the classification of healthy subjects and patients with LE using state-of-the-art features shows that only the classification task between healthy and LE patients is quite challenging in the same model and with the same signals. Using state-of-the-art features will not provide us with a powerful tool for differentiation between healthy subjects and patients. Nevertheless, for practical usage of this method and considering other similar diseases in the epicondyle region, it is beneficial to utilize this method after a physical examination, especially when the physician has doubts about the initial diagnosis and needs more information to diagnose lateral epicondylitis accurately. Moreover, our study exclusively enrolled male subjects, despite LE affecting both genders equally [2,10,11]. As a result, caution should be exercised when generalizing our findings across genders. Future studies should include both male and female participants to ensure more comprehensive and gender-inclusive conclusions. In addition, our study was limited to right-handed subjects, which may restrict the generalizability of our findings as LE can affect individuals irrespective of hand dominance.

This study has some limitations regarding the small sample size. Recording data for a larger number of subjects will increase the statistical power of the results. Although our analysis employed various validation strategies, such as leave-one-subject-out (LOSO) to prevent data leakage, rigorous statistical analysis, and random label testing, external validation on an independent dataset is still necessary to fully assess the model’s generalization capability. Another issue was the differentiation between healthy subjects and those with only LE. As mentioned above, other types of diseases can affect the epicondyle region, and it would be beneficial to conduct a study with a sufficient number of patients affected by any disease in the epicondyle region.

The dataset we used in this study is not suitable for investigating progress or improving the disease based on wearable devices.

### 4.8. Future Studies

For future studies, it would be beneficial to investigate PAC features in different muscles and assess the disease’s effect and progression in various patients’ muscles compared to healthy subjects. Additionally, it would be valuable to calculate PAC features in different neuromuscular disorders for further studies, compare the results, and uncover new information about these diseases.

To monitor the progress or improvement of the disease in patients using a wearable device, we need to record data suitable for such conditions. For further studies, recording data with wearable devices and then analyzing it to monitor the progress of the disease is a good aim.

## 5. Conclusions

In this study, we investigated the diagnosis of lateral epicondylitis using HD-sEMG signals. We introduced PAC features for HD-sEMG signals and demonstrated their robustness and superiority compared to state-of-the-art features and WT features. Our training involved a feedforward neural network utilizing PAC, WT, and state-of-the-art features to predict the labels of unseen subjects. The results revealed that PAC features performed better, correctly identifying all unseen subjects. These features might offer potential for investigating the pathological effects of various neuromuscular disorders and hold promise for diagnosing other neuromuscular diseases.

## Figures and Tables

**Figure 1 diagnostics-14-02255-f001:**
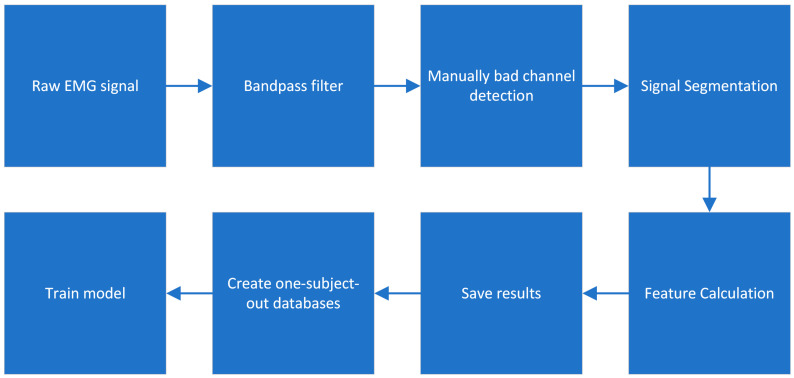
Pre-processing steps for preparing data for the training of the model.

**Figure 2 diagnostics-14-02255-f002:**
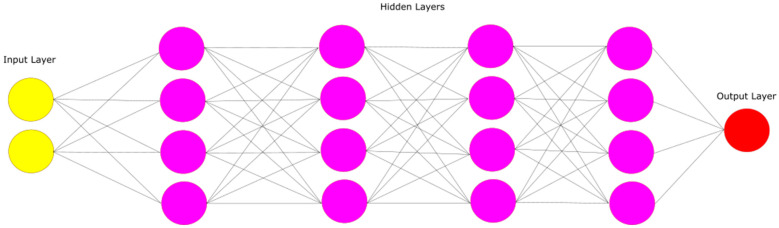
The proposed feedforward neural network aims for binary classification between patients and controls. The network consists of four hidden layers, each comprising one hundred and twenty-eight nodes and employing a rectified linear activation function. A sigmoid activation function was selected for the final layer, and the binary cross-entropy loss function was utilized during training.

**Figure 3 diagnostics-14-02255-f003:**
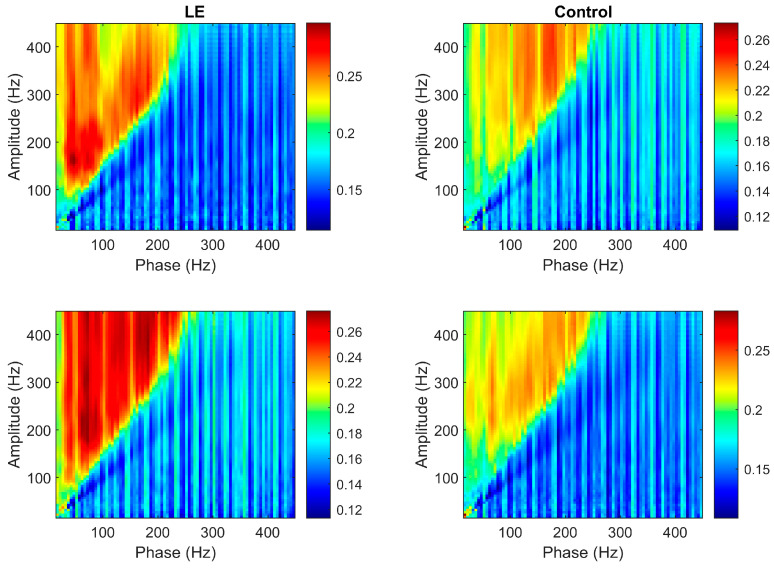
PAC features of two lateral epicondylitis (LE) patients (**left**) and two healthy controls (**right**). The patients show stronger coupling in lower frequency bands (50–250 Hz), with higher intensity in the phase–amplitude coupling, particularly in the 100–200 Hz range. In contrast, the healthy controls demonstrate a more evenly distributed coupling across frequencies, with lower overall intensity. The concentrated coupling in the patients’ lower frequency bands suggests abnormal neural activity, while the more balanced pattern in the healthy controls reflects typical neural communication.

**Figure 4 diagnostics-14-02255-f004:**
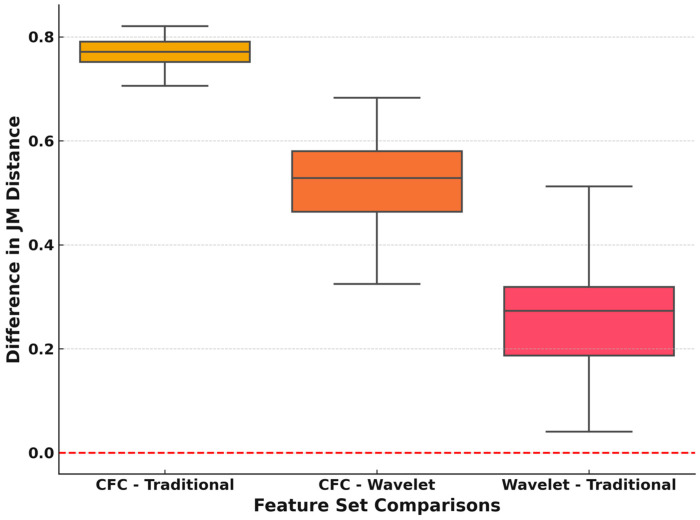
Boxplot of pairwise comparisons of JM distances among CFC, wavelet, and traditional feature sets.

**Figure 5 diagnostics-14-02255-f005:**
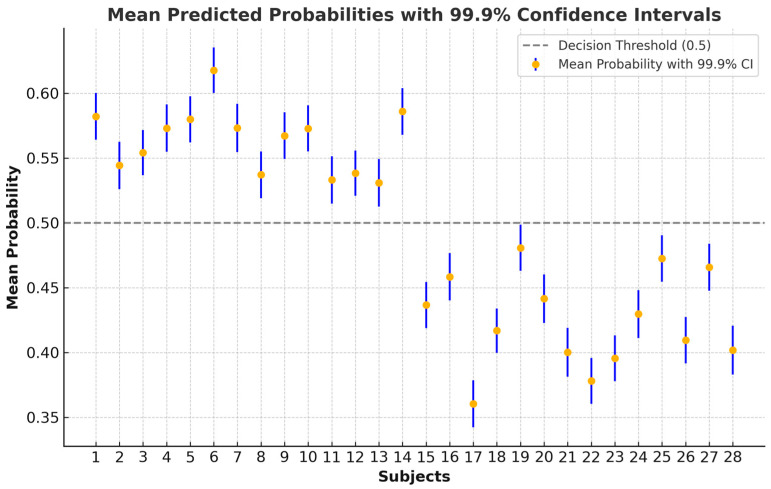
Mean predicted probabilities with 99.9% confidence intervals for the more than 4800 analyzed epochs of each subject (subjects 1–14: case; 15–28: control).

**Figure 6 diagnostics-14-02255-f006:**
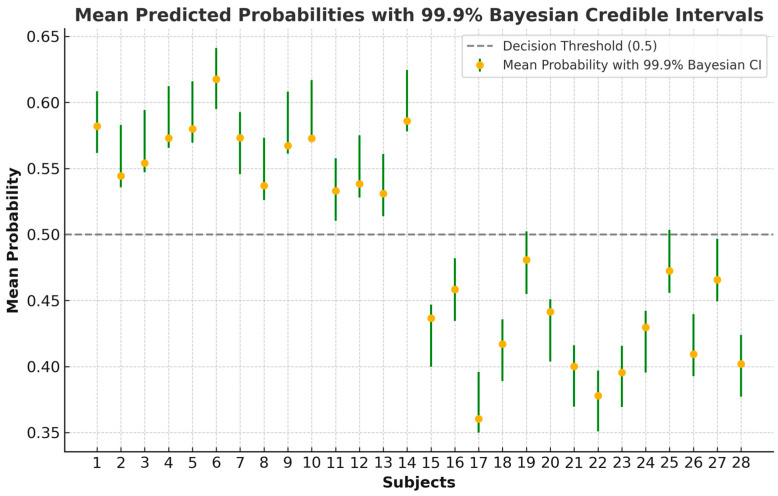
Mean predicted probabilities with 99.9% Bayesian credible intervals for the more than 4800 analyzed epochs of each subject (subjects 1–14: case; 15–28: control).

**Table 1 diagnostics-14-02255-t001:** The performance of the PAC features in each one-subject-out dataset (leave-one-subject-out validation).

Dataset ^†^	Training Accuracy	Validation Accuracy	Label of Unseen Subject
p1 out dataset	94.55	98.38	Case
p2 out dataset	94.35	98.32	Case
p3 out dataset	94.21	98.39	Case
p4 out dataset	94.45	98.68	Case
p5 out dataset	94.29	98.31	Case
p6 out dataset	94.21	98.26	Case
p7 out dataset	94.53	98.66	Case
p8 out dataset	94.37	98.37	Case
p9 out dataset	94.41	98.25	Case
p10 out dataset	94.28	98.44	Case
p11 out dataset	94.48	98.67	Case
p12 out dataset	94.38	98.34	Case
p13 out dataset	94.55	98.59	Case
p14 out dataset	94.46	98.30	Case
s1 out dataset	94.51	98.55	Control
s2 out dataset	94.29	98.34	Control
s3 out dataset	94.30	98.56	Control
s4 out dataset	94.17	98.47	Control
s5 out dataset	94.14	98.08	Control
s6 out dataset	94.36	98.43	Control
s7 out dataset	94.57	98.62	Control
s8 out dataset	94.17	98.53	Control
s9 out dataset	94.25	98.50	Control
s10 out dataset	94.32	98.46	Control
s11 out dataset	94.22	98.25	Control
s12 out dataset	94.29	98.40	Control
s13 out dataset	94.45	98.35	Control
s14 out dataset	94.23	98.59	Control
Average	94.34± 0.12	98.43±0.14	-

^†^ patients with the prefix “p” and healthy controls with the prefix “s”.

**Table 2 diagnostics-14-02255-t002:** The performance of the state-of-the-art features (excluding WT) in one-subject-out datasets (leave-one-subject-out validation).

Dataset ^†^	Training Accuracy	Validation Accuracy	Label of Unseen Subject ^††^
p1 out dataset	58.57	57.73	Control
p2 out dataset	58.13	58.22	Control
p3 out dataset	59.07	60.06	Control
p4 out dataset	58.20	58.74	Case
p5 out dataset	58.91	59.14	Control
p6 out dataset	58.13	58.89	Case
p7 out dataset	59.21	60.19	Control
p8 out dataset	58.91	57.60	Case
p9 out dataset	59.01	59.40	Control
p10 out dataset	57.87	56.66	Case
p11 out dataset	58.86	59.45	Control
p12 out dataset	57.54	58.70	Case
p13 out dataset	52.93	53.32	Control
p14 out dataset	59.81	61.01	Control
s1 out dataset	58.19	59.23	Case
s2 out dataset	58.97	60.04	Case
s3 out dataset	58.94	59.42	Case
s4 out dataset	58.85	59.80	Case
s5 out dataset	58.11	59.15	Control
s6 out dataset	57.48	58.10	Control
s7 out dataset	58.73	58.35	Case
s8 out dataset	57.38	59.30	Control
s9 out dataset	58.23	58.69	Control
s10 out dataset	59.54	59.63	Case
s11 out dataset	58.14	59.05	Case
s12 out dataset	58.47	58.97	Case
s13 out dataset	58.21	59.43	Case
s14 out dataset	58.67	59.47	Case
Average	58.32±1.21	58.84±1.39	-

^†^ patients with the prefix “p” and healthy controls with the prefix “s”; ^††^ erroneous classifications were underlined.

**Table 3 diagnostics-14-02255-t003:** The performance of the wavelet features in one-subject-out datasets (leave-one-subject-out validation).

Dataset ^†^	Training Accuracy	Validation Accuracy	Label of Unseen Subject ^††^
p1 out dataset	81.54	86.32	Case
p2 out dataset	81.48	86.08	Case
p3 out dataset	81.26	85.87	Case
p4 out dataset	81.51	85.92	Case
p5 out dataset	81.49	86.14	Control
p6 out dataset	81.27	86.00	Case
p7 out dataset	81.70	86.52	Case
p8 out dataset	81.57	86.34	Case
p9 out dataset	81.36	85.47	Control
p10 out dataset	81.26	85.62	Case
p11 out dataset	81.01	85.47	Control
p12 out dataset	81.37	86.49	Case
p13 out dataset	81.73	86.46	Case
p14 out dataset	81.36	85.66	Case
s1 out dataset	81.67	86.38	Control
s2 out dataset	81.23	85.64	Control
s3 out dataset	81.30	86.17	Control
s4 out dataset	81.44	85.95	Case
s5 out dataset	81.27	86.36	Control
s6 out dataset	81.29	86.37	Control
s7 out dataset	81.44	85.81	Case
s8 out dataset	81.72	85.90	Control
s9 out dataset	81.30	86.23	Control
s10 out dataset	81.40	86.20	Control
s11 out dataset	81.05	85.70	Control
s12 out dataset	81.11	85.76	Control
s13 out dataset	81.28	86.12	Control
s14 out dataset	81.44	85.78	Control
Average	81.38±0.18	86.02±0.31	-

^†^ patients with the prefix “p” and healthy controls with the prefix “s”; ^††^ erroneous classifications were underlined.

**Table 4 diagnostics-14-02255-t004:** The results of the JM distance for PAC features, state-of-the-art features, and wavelet features.

Dataset ^†^	PAC	State-of-the-Art Features (without Wavelet)	Wavelet (db4)
p1 out dataset	1.79	0.52	1.30
p2 out dataset	1.79	0.53	1.30
p3 out dataset	1.79	0.51	1.31
p4 out dataset	1.81	0.52	1.32
p5 out dataset	1.82	0.57	1.31
p6 out dataset	1.75	0.51	1.32
p7 out dataset	1.83	0.54	1.32
p8 out dataset	1.84	0.52	1.30
p9 out dataset	1.80	0.51	1.30
p10 out dataset	1.81	0.53	1.31
p11 out dataset	1.82	0.51	1.33
p12 out dataset	1.81	0.51	1.26
p13 out dataset	1.82	0.53	1.34
p14 out dataset	1.81	0.51	1.34
s1 out dataset	1.80	0.39	1.31
s2 out dataset	1.80	0.54	1.32
s3 out dataset	1.80	0.53	1.25
s4 out dataset	1.82	0.54	1.31
s5 out dataset	1.80	0.55	1.31
s6 out dataset	1.76	0.54	1.35
s7 out dataset	1.81	0.53	1.30
s8 out dataset	1.80	0.55	1.30
s9 out dataset	1.85	0.34	1.32
s10 out dataset	1.82	0.56	1.31
s11 out dataset	1.75	0.54	1.27
s12 out dataset	1.81	0.56	1.31
s13 out dataset	1.80	0.54	1.30
s14 out dataset	1.84	0.88	1.35
Average	1.80±0.02	0.53±0.08	1.30±0.02

^†^ patients with the prefix “p” and healthy controls with the prefix “s”.

**Table 5 diagnostics-14-02255-t005:** The performance assessment of different diagnosis models trained on different muscles (leave-one-subject-out validation).

Model	Sensitivity	Specificity	Accuracy
PAC all muscles	100	100	100
PAC ECR muscle	100	100	100
PAC ECU muscle	92.86 [79.37, 100]	100	96.43 [89.68, 100]
PAC EDC muscle	100	100	100
PAC FCR muscle	92.86 [79.37, 100]	100	96.43 [89.68, 100]
State-of-the-art all muscles	35.71 [10.61, 68.78]	28.57 [4.90, 52.23]	32.14 [7.75, 56.52]
State-of-the-art ECR muscle	42.86 [16.93, 68.78]	64.29 [39.18, 89.38]	53.57 [28.05, 56.52]
State-of-the-art ECU muscle	28.57 [4.90, 52.23]	50 [23.80, 76.19]	39.29 [14.35, 63.21]
State-of-the-art EDC muscle	7.14 [0, 20.63]	0	3.57 [0, 10.31]
State-of-the-art FCR muscle	50 [23.80, 76.19]	35.71 [10.61, 60.81]	42.86 [19.70, 68.5]
Wavelet all muscles	78.57 [63.37, 93.77]	85.71 [72.75, 98.67]	82.14 [67.95, 96.33]
Wavelet ECR muscle	78.57 [63.37, 93.77]	71.43 [54.70, 88.16]	75.00 [58.96, 91.04]
Wavelet ECU muscle	64.29 [46.54, 82.04]	71.43 [54.70, 88.16]	67.86 [50.56, 85.16]
Wavelet EDC muscle	50.00 [31.48, 68.52]	85.71 [72.75, 98.67]	67.86 [50.56, 85.16]
Wavelet FCR muscle	78.57 [63.37, 93.77]	78.57 [63.37, 93.77]	78.57 [63.37, 93.77]

## Data Availability

The data are not publicly available due to confidentiality agreements and privacy concerns but can be accessed upon reasonable request to ensure proper use and adherence to ethical guidelines. Additionally, we have provided a representative dataset.

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
