# Peer review of "AML-DECODER: Advanced Machine Learning for HD-sEMG Signal Classification—Decoding Lateral Epicondylitis in Forearm Muscles"

_diagnostics, 2024, doi:10.3390/diagnostics14202255_

Round 1

Reviewer 1 Report

Comments and Suggestions for Authors

The abstract mentions achieving "100% specificity and sensitivity" with PAC features, which may seem overly optimistic and could be questioned regarding potential overfitting, especially given the small sample size.

The article mentions a sample size of 14 patients and 14 controls, determined by a significance level of 0.05 and a power of 0.8. However, this small sample size might limit the generalizability of the findings. The justification for this sample size should be more robust, especially when claiming high predictive performance.

The explanation of the CFC and PAC calculations is overly technical and might confuse readers who are not specialists in the field. Simplifying or providing more intuitive explanations could enhance accessibility.

The study is limited to male, right-handed participants, which reduces the external validity of the findings. The authors should discuss how this limitation might affect the applicability of their results to other populations.

The perfect accuracy reported for the PAC-based model raises concerns about potential overfitting, especially with such a small dataset. This should be addressed, and more rigorous validation techniques, like cross-validation, should be considered.

The reporting of the statistical methods and results lacks detail. For instance, confidence intervals and effect sizes for the main findings are not adequately discussed, which are crucial for interpreting the significance and robustness of the results.

Comments on the Quality of English Language

minor editing

Reviewer 2 Report

Comments and Suggestions for Authors

The study focuses on wearable devices and high-density surface electromyography (HD-sEMG), in line with the growing use of portable, non-invasive technologies for continuous monitoring and diagnostics. These results point toward the new potential of at-home or real-time disease monitoring, with outstanding performance through the use of PAC features: 100% sensitivity and specificity in diagnosing lateral epicondylitis in unseen subjects. This means that features of PAC are very reliable in the context of neuromuscular disease diagnosis and bring about significant improvement over state-of-the-art methods. However, comments below, if suitably addressed, provide an opportunity for this study to be accepted in your journal:

  1. The current study included only 14 healthy and 14 patients with lateral epicondylitis. This could be considered a relatively small dataset. This, however, may restrict generalizability of the findings, and further validation on a more heterogeneous population is warranted.
  2. The study clearly showed high diagnostic efficiency in diagnosing lateral epicondylitis; however, a PAC feature application in other neuromuscular diseases would be purely hypothetical. More research generalizing the method to other disorders needs to be done.
  3. This study is likely to have an internal validation, thus not reflecting the reality of the situation. External validation using an independent dataset would be needed in order for this PAC-based neural network model to prove its robustness and generalizability.
  4. The study reports superior results for the features derived from a PAC but does not present a full comparison with many other established diagnostic features or methods. However, the lack of comprehensive benchmarking with other methods could constrain the insight into how PAC features perform in comparison to the rest of the field at large.
  5. The achievement of 100% sensitivity and specificity brings with it the worry that overfitting will take place—especially in a small sample. Overfitting may imply that the model is not generalizable enough to new, unseen data and to larger, more heterogeneous populations.
  6. Although these features of the PAC in their study gave great results, chances are that other multimodal approaches could be applied to get even better accuracies. By only focusing on one type of feature—a PAC—the study may lose an opportunity to develop more hybrid models that involve more features for improved diagnostic accuracy across the contexts.

Round 2

Reviewer 1 Report

Comments and Suggestions for Authors

The article is sufficiently developed and can be published.

Reviewer 2 Report

Comments and Suggestions for Authors

I am not qualified to assess the quality of English in this paper.